# Deep Learning-Assisted High-Throughput Analysis of Freeze-Fracture Replica Images Applied to Glutamate Receptors and Calcium Channels at Hippocampal Synapses

**DOI:** 10.3390/ijms21186737

**Published:** 2020-09-14

**Authors:** David Kleindienst, Jacqueline Montanaro, Pradeep Bhandari, Matthew J. Case, Yugo Fukazawa, Ryuichi Shigemoto

**Affiliations:** 1Institute of Science and Technology (IST )Austria, 3400 Klosterneuburg, Austria; jacqueline.montanaro@ist.ac.at (J.M.); pradeep.bhandari@ist.ac.at (P.B.); matthew.case@ist.ac.at (M.J.C.); 2Department of Histological and Physiological Sciences, Faculty of Medical Science, University of Fukui, Yoshida, Fukui 910-1193, Japan; yugo@u-fukui.ac.jp

**Keywords:** electron microscopy, SDS-digested freeze–fracture replica labeling, image analysis, glutamate receptors, voltage-gated calcium channels, CA1, stratum oriens, stratum radiatum, deep learning

## Abstract

The molecular anatomy of synapses defines their characteristics in transmission and plasticity. Precise measurements of the number and distribution of synaptic proteins are important for our understanding of synapse heterogeneity within and between brain regions. Freeze–fracture replica immunogold electron microscopy enables us to analyze them quantitatively on a two-dimensional membrane surface. Here, we introduce Darea software, which utilizes deep learning for analysis of replica images and demonstrate its usefulness for quick measurements of the pre- and postsynaptic areas, density and distribution of gold particles at synapses in a reproducible manner. We used Darea for comparing glutamate receptor and calcium channel distributions between hippocampal CA3-CA1 spine synapses on apical and basal dendrites, which differ in signaling pathways involved in synaptic plasticity. We found that apical synapses express a higher density of α-amino-3-hydroxy-5-methyl-4-isoxazolepropionic acid (AMPA) receptors and a stronger increase of AMPA receptors with synaptic size, while basal synapses show a larger increase in N-methyl-D-aspartate (NMDA) receptors with size. Interestingly, AMPA and NMDA receptors are segregated within postsynaptic sites and negatively correlated in density among both apical and basal synapses. In the presynaptic sites, Cav2.1 voltage-gated calcium channels show similar densities in apical and basal synapses with distributions consistent with an exclusion zone model of calcium channel-release site topography.

## 1. Introduction

The molecular anatomy of the synapse, which encompasses the composition and arrangement of ion-channels, the vesicle-release machinery, neurotransmitter receptors, scaffolding proteins, intracellular signal transduction molecules and cell-adhesion molecules within and close to the pre- and postsynaptic membrane specializations, determines the characteristics of neurotransmission and influences synaptic plasticity [1,2,3,4,5,6]. On the presynaptic side, the distance between voltage-gated Ca^2+^ channels and Ca^2+^ sensors of exocytosis influences the speed and efficacy of neurotransmitter release [4]. On the postsynaptic side, a higher density of postsynaptic receptors causes a larger excitatory postsynaptic current (EPSC) when exposed to the same neurotransmitter concentration [7]. Not only does the number and density of receptors matter, but also the position and spatial relationship between synaptic proteins: Receptors near the vesicle fusion site will be exposed to higher local neurotransmitter concentrations, which triggers a larger postsynaptic potential [1,7,8]. Evoked transmission preferentially occurs at sites of Rab3-interacting molecule (RIM) nanoclusters and these nanoclusters are reported to align with postsynaptic α-amino-3-hydroxy-5-methyl-4-isoxazolepropionic acid (AMPA) receptors (AMPAR) in hippocampal neurons [9]. Furthermore, the subunit composition of postsynaptic receptors and their association with auxiliary subunits shape the postsynaptic response [2,10,11,12] and affect plasticity [2,12,13,14,15,16]. Understanding how the synaptic molecular anatomy affects neurotransmission, how it differs between synaptic connections to support functional differences between different types of synapses and how it contributes to plasticity and is affected by diseases are central challenges in molecular neuroscience.

One of the best-studied synaptic connections is the one between hippocampal CA3 and CA1 pyramidal neurons; the majority of studies have focused on synapses onto spines of apical dendrites (trunk and oblique branch) in stratum radiatum (SR) while fewer have investigated synapses onto spines of basal dendrites in stratum oriens (SO). Excitatory inputs in SR originate almost exclusively in CA3, while basal dendrites receive major projections from CA3 and CA2, with CA3 accounting for more than 80% of these projections, and minor projections from the entorhinal cortex, amygdala as well as recurrent collaterals from CA1 pyramidal neurons [17,18]. The projection pattern to CA1 is not uniform within the CA3, with proximal and distal CA3 neurons preferentially projecting to SR and SO, respectively [19]. CA3-CA1 synapses in SR and SO differ in their intra- and extracellular signaling cascades involved in plasticity [20,21] and long-term potentiation (LTP) is more readily induced and/or more strongly expressed in SO than in SR [20,21,22,23]. Furthermore, synaptic plasticity in SR and SO is differentially modulated by dopamine [21,23,24] and acetylcholine [25]. Interestingly, glutamate receptor subunit allocation shows a left–right asymmetry with synapses in SR and SO being affected in an opposite manner [26,27,28]. Given the clear differences between these seemingly very similar synaptic connections, we wondered whether postsynaptic glutamate receptors and presynaptic voltage-gated calcium channels show differential distributions in these layers.

SDS-digested freeze–fracture replica labeling (SDS-FRL) electron microscopy allows visualization of the membrane and membrane-anchored proteins with high resolution and sensitivity [29,30,31]. However, detailed image analysis is very time-consuming because the accurate demarcation of synaptic areas, as well as annotation of gold particles, is necessary. We have previously increased the speed of analysis, by automating the gold particle detection [32], but the manual demarcation of the synaptic membrane specialization still remains the most time-consuming part of a typical SDS-FRL experiment.

The last decade has seen great progress in the field of machine learning: Deep learning algorithms have reached or surpassed human-level accuracy in classification tasks [33,34], and have greatly improved performance in image segmentation tasks [35]. They are increasingly used for biomedical applications, from tumor segmentation [36,37] to neuron segmentation in calcium imaging experiments [38]. In EM-image analysis, deep learning has been applied to, for example, mitochondria segmentation [39], synaptic cleft segmentation [40] and connectomics [41,42,43]. Harboring the power of deep learning for image analysis promises to speed up analysis and increase reproducibility. The development of user-friendly software tools whose use does not require programming experience will be necessary for the widespread adoption of automated image analysis methods.

In this paper, we report the development of Darea (Deep learning-assisted replica image analysis suite), a software which utilizes deep learning for rapid demarcation of synaptic membrane compartments of SDS-FRL images, combined with the GPDQ software [32] we previously developed for automated gold particle detection. Darea can be used for subsequent analysis of particle distributions within synapses and comparison of real data with Monte-Carlo simulations to characterize specific distribution patterns. We proceeded to apply this software to investigate the distribution of AMPAR, N-methyl-D-aspartate (NMDA) receptor (NMDAR) and Cav2.1 subunit of P/Q-type voltage-dependent calcium channels distributions in synapses on apical and basal dendritic spines of mouse CA1 pyramidal neurons.

## 2. Results

### 2.1. Darea Software

We developed Darea software from the idea that deep learning methods should assist the user to allow for high-throughput analysis of replica images and readily enable review and, if necessary, correction of automatically generated demarcations. Darea builds on the previously developed GPDQ software [32] and therefore also features automated and manual gold particle detection previously developed, quantification of labeling density, estimation of nearest-neighbor distances (NNDs) and Monte-Carlo simulations for characterizing distribution patterns such as clustering and co-localization. Darea offers a straight-forward graphical interface that allows users to readily use their images for training demarcation and particle detection algorithms. In addition, it offers an intuitive tool for visualizing analyzed images.

### 2.2. Automated Demarcation of Synaptic Membranes and Annotation of Gold Particles

We trained Darea on a dataset of 1148 replica images of demarcated intramembrane particle (IMP) clusters on the E-face, which correspond to postsynaptic density (PSD), from CA1 SR and SO for PSD segmentation, and on a dataset of 1090 images of demarcated presynaptic active zones (AZs) from interpeduncular nucleus (IPN) for AZ segmentation [44]. The AZ demarcation on replica images has been verified with its area distribution, which had no significant difference from that of PSD in the same replica samples (Figure 4G in [44]). The manual correction was to revise machine-generated (Darea) demarcations with mostly minor changes, but major revision was necessary for a few Darea demarcations (examples shown in Appendix A). For PSD demarcation (Figure 1A,B), medians of intersect over union (IoU, see methods) between Darea or corrected Darea and human demarcations were 83 or 84%, respectively (Figure 1C). These values are higher than IoU between demarcations made by the person who provided the training data (Person 1) and a second person (Person 2) (72%, Figure 1C), indicating high reliability of the machine-generated demarcation. The median area of demarcated PSD by Darea or corrected Darea was 99.7% or 94.2%, respectively, of that by Person 1, while that by Person 2 was 79.4% (Figure 1D). For AZ demarcation (Figure 1E,F), medians of IoU between Darea or corrected Darea and human demarcations were 66% or 79%, respectively (Figure 1G). The median area of demarcated AZ by Darea or corrected Darea was 112% or 101%, respectively, of that by human (Figure 1H). The IoU for AZ shows larger deviation than that for PSD (Figure 1C,G), and the distribution of AZ area shows larger spread than that of PSD area (Figure 1D,H). This might be explained by the fact that the AZ training dataset was from a different brain area (IPN). Subtle differences in morphology not reflected in the training dataset are expected to cause errors in prediction. Nonetheless, it is quite reassuring that demarcations are reasonably accurate even when trained on a different brain region, indicating that the same well-trained network may be used for automated synapse demarcation in many brain regions. For results reported in later figures the corrected Darea dataset was used and further samples were analyzed in the same way, i.e., by automated demarcation followed by manual review and correction.

Annotation of gold particles within the demarcated region of interest is the next step in a typical analysis workflow. We have previously reported automated gold particle detection on replica images using a finding circles algorithm followed by a naïve Bayes classifier [32]. Here, we compare the detection accuracy of this naïve Bayes classifier and a random forest classifier (Appendix A). We find that random forest tends to have a lower false-positive rate and a larger false-negative rate than naïve Bayes. Random forest is more sensitive in distinguishing different gold particle sizes, while naïve Bayes sometimes assigns particles to both classes (Appendix A). We generally recommend using a random forest classifier, especially for double labelings. Naïve Bayes works well on single labelings if the secondary antibody, like the one used for our Cav2.1 labelings, has a wide range of gold particle sizes.

### 2.3. Glutamate Receptor Distributions in CA1 Pyramidal Cell Spine Synapses

We conducted double labelings using an antibody against the obligatory GluN1 subunit of NMDAR and an antibody reactive to GluA1-3 subunits of AMPAR [45] on replicas containing the CA1 area of the dorsal hippocampus. AMPAR expression in SR shows a gradient with synapses close to the pyramidal layer expressing fewer AMPAR than distal synapses [46]. We, therefore, limited our analysis to spine synapses in the middle one-third of SR and those in SO. We used Darea for automatically demarcating the PSD and detecting gold particles within or less than 30 nm from the demarcation border (outer rim) and manually corrected its results when appropriate. We included this outer rim because gold particle centers may be up to 30 nm distant from the epitope because of the sizes of both primary and secondary antibodies (~10 nm each [47]), as well as the gold particles themselves. However, the outer rim may also contain particles representing extrasynaptic receptors. Thus, we counted numbers of AMPAR particles contained in 10–60 nm-width outer rims (Appendix A), and found that the number linearly increases up to 30 nm, showing a less steep but stable increase after 40 nm. This result indicates that AMPAR particles within the 30 nm-width outer rim mostly represent synaptic receptors. With this definition of particles for synaptic receptors, we found a higher labeling density for AMPAR in SR than SO (*p* = 0.02), while GluN1 labeling density was similar in both strata (*p* = 0.5) (Figure 2A). For detailed test statistics of this and following results see Appendix A.

It has previously been reported that NMDARs are more centrally localized than AMPARs [48,49,50,51]. In our results, we found about one-quarter of AMPAR but only about one-eighth of GluN1 particles localized in the outer rim (AMPAR: SR: 23.7%, SO: 24.5%; GluN1: SR: 12.4%, SO: 10%). To examine the subsynaptic distribution more quantitatively, we calculated a center-periphery-index (CPI, see methods). A CPI of 0 indicates localization at the center of gravity, a CPI of 1 indicates localization on the demarcation border, while a CPI > 1 indicates localization in the outer rim. We found AMPAR to be more peripherally localized than GluN1 in both SR and SO (Figure 2C) (SR: *p* = 0.0007, SO: *p* = 0.0001). This may indicate peripheral localization of AMPAR, central localization of GluN1 or both. To distinguish these possibilities, we conducted Monte-Carlo simulations of randomly distributed particles (Figure 2B). To simulate particles including those in the outer rim properly, we used a two-step approach: We first simulate the location of the epitope within the demarcated synaptic areas, giving each pixel the same probability of epitope location while keeping a minimum distance of 6 nm (assumed from the radius of these channels). In the 2nd step, the simulated particle is placed at a random distance (0–30 nm) from the epitope at a random angle. Simulated gold particles cannot be closer than 10 nm (based on the size of particles) to each other or to real gold particles of the other size. About 10% of simulated particles were found in the outer rim (AMPAR: SR: 9.6%, SO: 9.9%; GluN1: SR: 9.5%, SO: 9.5%), similar to the real GluN1 particles (SR: 12.4%, SO: 10%). Comparing the CPI for real and simulated AMPAR and GluN1 particles (Figure 2C) we conclude that AMPARs are localized significantly more peripheral than simulated particles (SR: *p* = 0.029, SO: *p* = 0.0059) while NMDARs show no preference for either periphery or center (SR: *p* = 0.97, SO: *p* = 0.75) in both strata.

We then examined whether AMPA and NMDA receptors show clustered distribution within the PSD, we therefore measured nearest-neighbor distances (NNDs) between 6 or 12 nm gold particles and compared them with NND distances of Monte-Carlo simulated gold particles (Figure 2D). We found NNDs between AMPAR significantly smaller than those expected by chance in SO (*p* = 0.0004) while the difference did not reach the significance level in SR (*p* = 0.13). NNDs between GluN1 particles did not differ from simulations (SR: *p* = 0.4, SO: *p* = 0.13). We next assessed the relationship of glutamate receptor number with PSD area and found clear positive correlations for both AMPAR and GluN1 in both strata (Figure 2E,F), with AMPAR showing a steeper correlation in SR than SO (*p* = 0.0002), while GluN1 showed a steeper correlation in SO (*p* = 0.031).

We next assessed how AMPAR and NMDAR are distributed with respect to each other. To our surprise, we found a significant negative correlation between AMPA and NMDA receptor density in both strata (Figure 3A). We then examined whether AMPAR and NMDAR show subsynaptic colocalization. To this end, we measured NND from AMPAR to GluN1 particles and compared them with those of virtual GluN1 particles generated by fitted Monte-Carlo simulations [32], which preserve the distances between GluN1 particles. We found AMPAR and GluN1 to be significantly more segregated than expected by chance in both strata (SR: *p* < 0.0001, SO: *p* = 0.0002) (Figure 3B). 

It is possible that a subset of synapses in either stratum may show different properties from the majority. For example, it would be easily conceivable that synapses in SO receiving input from CA2 show clear differences in glutamate receptor distributions from CA3 input synapses in either SR or SO. However, since these synapses are comparatively infrequent, these differences might not be reflected in our analysis. To test such a possibility, for synapse showing labeling for both GluN1 and AMPAR, we combined all the variables we measured (PSD area, AMPAR and GluN1 particle number, NNDs between particles, CPI), using a dimensionality reduction technique, into a *t*-distributed stochastic neighbor embedding (*t*SNE) plot (Figure 3C). SR and SO synapses were intermingled in this plot, indicating that individual synapses are not fundamentally different in their glutamate receptor distributions in these two strata. Including synapses without labeling for either AMPAR or GluN1 leads to the identification of two additional clusters of AMPAR or GluN1 negative synapses in this plot (not shown) but did not change the conclusion. Given that CA2 synapses occupy about 20% of spine synapses in SO, we think it is reasonable to assume that our SO sample (*n* = 91) includes a relevant number of CA2 input synapses for this analysis, while synapses receiving inputs from other brain regions may be too rare. If so, we may conclude that individual CA2-CA1 synapses do not show a fundamentally different glutamate receptor markup from CA3-CA1 synapses, though this result does not exclude potential differences on the population level.

High-resolution antibody double labelings always carry the risk of steric hindrance between antibodies masking potential epitopes and even more so when antibodies are conjugated to bulky gold particles. In all our experiments we’ve applied different primary and secondary antibodies together to distribute this risk at least equally among the two antibodies. However, it is possible that the masking effects of secondary antibodies differ depending on the size of the conjugated gold particle. Therefore, we repeated double-labeling experiments with switched gold particle sizes (i.e., 12 nm for AMPAR and 6 nm for GluN1) (Appendix A). We observed that 6 nm secondary antibody had about twice the labeling efficiency of 12 nm secondary antibody (Figure 2A and Appendix A). We could mostly reproduce our previous results, with a few notable exceptions: First, the difference in AMPAR density between SR and SO lost its significance (Appendix A, *p* = 0.13). Second, the difference in the regression line of AMPAR receptor number and PSD area between SR and SO became not significant (Appendix A, *p* = 0.3). Third, the clustering feature of AMPAR observed in SO (Figure 2D) was also lost (Appendix A). These discrepancies may be due to the lower labeling efficiency for AMPAR with 12 nm secondary antibody, hampering the difference detection. To test this possibility, we performed single labeling for AMPAR using a 5 nm secondary antibody which showed a higher labeling efficiency (353 particles/µm^2^, Appendix A) than those with the 6 nm (83 particles/µm^2^, Figure 2) and 12 nm (52 particles/µm^2^, Appendix A) secondary antibodies. We found AMPAR particle density significantly higher in SR than SO (Appendix A, *p* = 0.025) and the regression line between PSD area and AMPAR particle number significantly steeper in SR (Appendix A, *p* < 0.0001), reproducing our results in Figure 2. The NNDs of AMPAR particles were also significantly smaller than simulated ones in SO (*p* = 0.0032, Appendix A). These results indicate that the discrepancies were indeed due to the lower sensitivity of the 12-nm secondary antibody. 

The existence of silent synapses, showing an NMDAR but not an AMPAR component in electrophysiological recordings, in the CA1 has been a matter of debate [52,53,54,55]. We examined 5 nm gold labeling for AMPAR on morphologically identified PSDs and found that less than 2% of the synapses observed (4 out of 235) showed an absence of AMPAR labeling. All of these four PSDs had partially fractured synaptic areas and AMPAR expression at these synapses can therefore not be excluded. This result supports the argument that no significant proportion of silent synapses exists in the CA1 area, at least in adult mice.

In conclusion, we found a higher density of AMPAR as well as a steeper correlation of AMPAR number with PSD area in SR than SO, a steeper correlation of GluN1 with PSD area in SO and a negative correlation of AMPAR and NMDAR densities, with those receptors being segregated within the PSD in both strata.

### 2.4. Cav2.1 Channel Distribution in SR and SO

Cav2.1 and Cav2.2 are the main calcium channels involved in neurotransmitter release at CA3 to CA1 synapses [56,57,58]. We assessed whether Cav2.1 shows differential distributions between spine synapses in SR and SO using SDS-FRL for Cav2.1. We verified our demarcations of AZs generated by Darea (corrected Darea), by comparing the area of complete AZs with the area of reconstructed PSDs obtained from serial ultrathin sections through the CA1 area (Appendix A), and observed a similar distribution of synaptic areas (Appendix A, *p* = 0.62). We found specific labeling for Cav2.1 in the presynaptic active zone with no difference in density between the two strata (Figure 4A, *p* = 0.54). We next investigated the relationship of Cav2.1 labeling with active zone size. We found a strong positive correlation between the number of Cav2.1 particles and active zone size (Figure 4B) in both layers (*p* < 0.0001 for both SR and SO). We then assessed the subsynaptic distribution of Cav2.1 in these layers and compared it to random distribution generated by Monte-Carlo simulations (Figure 4C). We first computed the mean CPI for real particles, and found it indistinguishable from that for simulated particles in both layers (*p* = 0.65, Appendix A), indicating that Cav2.1 shows no preferential localization to either periphery or center of the synapse.

We assessed whether Cav2.1 particles localize more closely to each other than expected by chance, and found that NNDs between real Cav2.1 particles are slightly, but significantly, smaller than those between simulated particles (Figure 4D). A few different models for the spatial relationship between Cav2.1 and release sites have been proposed [59,60,61]. In synapses displaying tight coupling between calcium influx and vesicle release sites a perimeter release model [59,60] and one-to-one correspondence of Cav2.1 clusters and vesicle release sites [61] have been proposed, while synapses with loose coupling were modeled by an exclusion zone model, with Cav2.1 being excluded from vesicle release sites but randomly distributed in the remaining synaptic area [59]. We tested whether our data of SR and SO synapses are consistent with either of these models: Cav2.1 clusters were defined as consisting of at least three particles with a distance of less than 39.6 nm (mean NND + 2 × SD) from each other (Figure 4C), similar to definitions used in previous literature [32,61]. We found significantly fewer Cav2.1 clusters (SR: *p* = 0.012; SO: *p* = 0.012) with larger number of particles (SR: *p* = 0.0004; SO: *p* = 0.0003) than predicted by simulations (Figure 4E,F). In synapses pooled from CA1 SR and stratum lacunosum moleculare, Schikorski and Stevens [62] observed 10.3 ± 5.6 docked vesicles per AZ, even more than the maximum number of Cav2.1 clusters (7) found in our data. Our data, therefore, do not support the one-to-one correspondence of Cav2.1 clusters and docked vesicles in spine synapses in CA1. We next tested if our data could be explained by the exclusion zone model. We estimated the number of exclusion zones with a formula $\left(1.4\;+\;226\times \mathrm{AZ}\;\mathrm{area}\;\left(\mathrm{in}\;{µm}^{2}\right)\right)$ based on Schikorski and Stevens’ data, Figure 4C in [62], and randomly distributed them within the AZ (see methods). We then performed two-step simulations of Cav2.1 particles; simulated Cav2.1 channels were excluded from the exclusion zones while simulated gold particles (up to 30 nm distant from simulated Cav2.1 channels) were allowed to be in the exclusion zones (Figure 4G). The mean NND between real Cav2.1 particles was not different from that of the simulated Cav2.1 particles (Figure 4H, SR: *p* = 0.29, SO: *p* = 0.19), indicating that our data are consistent with the exclusion zone model.

## 3. Discussion

In this study, we report the development of Darea, a convenient software tool for semi-automatic replica analysis. We show that Darea can automatically demarcate PSDs and AZs with very similar median size as human demarcation, with less variability than was observed between two human investigators. In our experience, Darea can facilitate the image analysis of replica samples by a factor of 5–10, depending on how much correction of the machine-generated demarcations is necessary. While we trained our networks for AZ and PSD demarcation by images with our EM configurations, it is possible that they will perform poorly on images taken with a different electron microscope or even the same microscope with different image acquisition settings. In this case, users can easily train the neural networks on their own images with Darea. For training their own networks, the use of at least 100 demarcated images would be recommended, but higher accuracy will be achieved with a larger number of training images. Darea is not limited to PSD and AZ demarcations but can be applied to any profiles with specific morphological features in freeze–fracture replica samples. Demarcation prediction can be performed on most CPUs within a reasonable time (<1 h for 100 images), but is at least an order of magnitude faster when run on a modern GPU. After manual review, automatically generated demarcations can be reused for further training the network, which is expected to lead to constant improvements in prediction accuracy. To our knowledge, Darea is the first software tool allowing users to utilize deep learning for their own replica EM image analysis using a graphical user interface that does not require prior knowledge about deep learning or programming.

We demonstrate the usefulness of Darea by applying it to analyses of replica labeling for AMPAR, NMDAR and Cav2.1 at spine synapses in CA1 SR and SO. Our analysis revealed several differences in glutamate receptor distributions between SR and SO synapses. Cav2.1 distributions, however, were remarkably similar. We observed a higher density of AMPAR in SR synapses, consistent with a previous study reporting higher synaptic efficacy in SR in vivo [23].

We observed a steeper correlation of AMPAR number with PSD area in SR and a steeper correlation of NMDAR number with PSD area in SO. This indicates that large SR synapses contain more AMPARs and fewer NMDARs than large SO synapses. It seems possible that large NMDAR-dense synapses in SO could participate in LTP more readily than large AMPAR-dense synapses in SR, which could, at least partially, explain the stronger LTP observed in SO [20,21]. The mechanistic differences in the pathways involved in LTP between SR and SO [20,21] may also be involved in generating this difference in correlation. For example, a difference in magnitude of activity-dependent GluA1 incorporation into potentiating synapses increasing their sizes and AMPAR density [27,63] could explain the steeper correlation of AMPAR in SR.

We found a negative correlation between AMPAR and NMDAR densities, which was unexpected because both NMDAR and AMPAR numbers positively correlate with the PSD area (Figure 2E,F), and LTP potentiates both the AMPAR and the NMDAR component [21,64]. This negative correlation is indicative of a wide spectrum of AMPAR/NMDAR ratios in CA1 synapses, which would not be detected by the recording of responses from many synapses. It is tempting to speculate that synapses with a high NMDAR and low AMPAR may be more plastic and thus better able to contribute to new memory encoding, whereas synapses with a low NMDAR and high AMPAR may serve for stable transmission. Future work is needed to understand the mechanism involved in generating this negative correlation as well as its physiological relevance.

We found AMPAR more peripherally distributed than NMDAR, consistent with previous studies [48,49,50,51]. However, these studies disagree whether this is due to the peripheral preference of AMPAR location, central preference of NMDAR location, or both. We compared our data with Monte-Carlo simulations of randomly distributed particles and found AMPAR located more peripheral than simulated particles while NMDAR was located similarly to simulated ones. This result is clearly inconsistent with a previous study [49] showing a central distribution of NMDAR at CA1 SR synapses. This difference may stem from the method used (post-embedding vs. SDS-FRL). One limitation of our study is, that the majority of PSDs are fractured only partially, which may compromise our conclusion. However, when we limit our analysis to only complete PSDs, we obtain similar CPI values, indicating that this is likely not confounding our result.

We have found AMPARs and NMDARs to be segregated within the PSD, in line with a recent study reaching the same conclusion using super-resolution imaging in hippocampal primary culture [51]. This result is consistent with the observation that AMPA and NMDA receptors are allocated to different receptor supercomplexes [65,66].

On the presynaptic side, we found Cav2.1 distribution in active zones to be remarkably similar between SR and SO. In both layers, Cav2.1 number correlated well with the AZ area, and Cav2.1 particles showed no preference for either central or peripheral localization. Cav2.1 particles are slightly closer to each other than expected by chance, a property that can be explained by the exclusion zone model of calcium channel-release site topography [59]. Although it is still unclear if the CA3-CA1 synapses have a loose or tight coupling of Cav channels and Ca^2+^ sensors [67,68], the exclusion zone model seems to apply better to more plastic synapses [59,69] consistent with our results. The visualization of docked vesicles and calcium channels in the same preparation would be necessary for direct measurement of coupling distance and verification of different topography models.

## 4. Materials and Methods 

### 4.1. Animals

Wild-type C57BL/6J (Jax, Bar Harbor, ME, USA; #000664) mice were initially purchased from Jackson Laboratory and were bred at the preclinical facility of IST Austria on a 12:12 light–dark cycle with access to food and water *ad libitum*. Experiments were performed on 2–3 months old male mice in the light phase of the cycle. All experiments were performed in accordance with the license approved by the Austrian Federal Ministry of Science and Research (Animal license number: BMWFW-66.018/0012-WF/V/3b/2016, approved on 19.06.2014) and the Austrian and EU animal laws.

### 4.2. Antibodies

All antibodies used in this study are described in Table 1.

### 4.3. SDS Freeze-Fracture Replica Labeling

SDS-FRL was performed, with modification, as described previously [31]: Mice were deeply anaesthetized with an overdose of Ketamine/Xylazine (660/13 mg/kg BM, i.p.) and perfused transcardially with 2% paraformaldehyde (PFA), 15% saturated picric acid in 0.1M PB, pH 7.3–7.4. The brain was extracted and 80 µm coronal slices were cut with a vibratome (linear slicer pro 7, Dosaka, Kyoto, Japan). Slices containing all layers of the CA1 of the dorsal hippocampus were cut out, cryoprotected in 30% Glycerol in PB and frozen using a HPM010 high pressure freezing machine (Bal-Tec, Balzers, Liechtenstein). They were fractured at −117 °C and coated with 5 nm carbon, 2 nm platinum-carbon and 20 nm carbon layers using a BAF060 freeze–fracture machine (Bal-Tec). Tissue was dissolved with SDS-solution (2.5% SDS, 20% Sucrose, 15 mM Tris, pH 8.3) shaking (50 rpm) for 18 h at 80 °C (AMPAR/GluN1 labeling) or for 48h at 60 °C (Cav2.1 labeling) in a hybridization incubator HB-100 (Taitec). Replicas were washed in SDS-Solution and 3x in washing buffer (WB; 0.1% Tween-20, 0.05% BSA (bovine serum albumin), 0.05% NaN_3_ in TBS (tris-buffered saline)), then incubated with blocking buffer (BB; 5% BSA in WB) for 30 min, followed by incubation with gp-anti-Cav2.1, rb-anti-GluA1-3 or mix of rb-anti-GluA1-3 and ms-anti-GluN1 diluted in BB for overnight at 15 °C. Replicas were washed 3x with WB, incubated with BB for 30 min, followed by incubation with secondary antibodies (gt-anti-gp-5 nm for Cav2.1, gt-anti-rb-5 nm for GluA1-3 or mix of gt-anti-rb-6 nm and gt-anti-ms-12 nm (Figure 2 and Figure 3) or mix of gt-anti-rb-12 nm and gt-anti-ms-6 nm (Appendix A) for GluA1-3/GluN1) diluted in BB overnight at 15 °C. Replicas were then washed 3x with WB, 1x with TBS and 2x with MilliQ and mounted on copper grids for EM observation. 

### 4.4. Imaging and Manual Demarcation

Replicas were imaged with a Tecnai 10 (FEI, 80kV accelerating voltage) or Tecnai 12 electron microscope (FEI, 120kV accelerating voltage). Images of E-face PSDs of spine synapses and P-face AZs putatively contacting spine synapse, identified by the shallow concave AZ structure, were taken from middle third of SR and from SO of the same replica at a similar medial-lateral position. The sample was tilted in order to maximize the synaptic membrane area on the image. Synapses containing at least 2 gold particles (irrespective of size) were included in the analysis, except when quantifying the percentage of synapses expressing AMPAR, where all morphologically identified PSDs were included in the analysis. When demarcating manually, we defined the PSD as area of tightly packed IMP clusters (distance between IMPs ≤ 15 nm). AZ was demarcated based on alteration in surface curvature and/or higher density of IMPs following previously reported practice [44]. Gold particles were counted if their center was inside or ≤ 30 nm away from the demarcation border.

### 4.5. Darea Software

Darea software is based on GPDQ software) [32], which we modified by adding a graphical interface for manual region of interest demarcation as well as the automated region of interest demarcation using deep learning (see below) based on Semantic Segmentation Suite [70]. Darea extends the Monte-Carlo simulations of GPDQ by adding the possibility for two-step simulations and simulations with exclusion zones (see below) as well as the gold particle detection by the addition of a random forest classifier as well as a graphical menu for training classifiers. Darea allows for the quantification of distances between particles and the center of gravity and edge of the demarcated area. Darea is implemented in Matlab (Mathworks) and python with TensorFlow [71]. Darea is open source software licensed under GPDL and can be found at https://github.com/DavidKleindienst/Darea.

#### 4.5.1. Deep Learning for Automated Segmentation of Synaptic Areas

For both training and automated segmentation, images are downsampled to a resolution of 512 × 512 pixels. In the case of rectangular images, the largest central square possible was first cropped and then downsampled. Two classes were used: background and PSD or AZ. 

In a preliminary analysis we compared several deep neural network architectures: SegNet [72], DenseASPP [73], DeepLabV3+ [74], FC-DenseNet56 [75] and FC-DenseNet103 [75]. Based on these results (not shown) we decided on FC-DenseNet103 architecture [75]. For training PSD and AZ demarcation, we used training datasets of 903 images of E-face PSDs from CA1 and 880 images of P-face AZs from interpeduncular nucleus and validation datasets of 245 images of E-face PSDs from CA1 and 210 images of P-face AZs from interpeduncular nucleus. Training was performed with data-augmentation (random horizontal and vertical flipping, random brightness alteration for up to ±50%) for 400 epochs on a pretrained network on an NVIDIA GTX1080Ti graphics card. We then chose the network after the epoch that had the highest accuracy on the validation dataset. The accuracy was measured as mean IoU, which was computed as the intersect of manual and machine demarcation divided by the union of these demarcations.

#### 4.5.2. Gold Particle Detection

We first detect putative gold particles in the image by using a standard circle detection algorithm. In total, 27 features (e.g., mean and standard deviation of pixel values of different regions of the circle) are computed and fed into a naïve Bayes [32] or random forest (16 trees) classifier, which then decides between particle and background. Classifiers were trained on 5 nm GluN2B and 10 nm GluN1 double-labeled images replica images from CA1 SR. Further, 5 and 10 nm trained classifiers can also be used for 6 and 12 nm particle detection, respectively (Appendix A).

#### 4.5.3. Monte-Carlo Simulations

Monte-Carlo simulations for generating random or fitted (keeping the NNDs between the simulated particles similar to the ones between the real particles) simulations were described previously [32]. Briefly, random x and y coordinates for a particle were chosen from a uniform distribution, with minimum and maximum corresponding to the demarcated region of interest, using Matlab’s in-built random number algorithm. Thus, each pixel of the demarcated region of interest has the same probability of becoming the center of a particle. A minimum distance of 10 nm between particle centers was kept by randomly redistributing all particles which were too close to another. However, such simulations performed on the demarcation only would underestimate the number of particles in the outer rim (0% in outer rim), while simulations including the outer rim would drastically overestimate the number of particles in the outer rim (~37% in outer rim) compared to real AMPAR (~24% in outer rim) or NMDAR (~11% in outer rim) particles. Here we modified these simulations using a two-step approach: We first simulate the location of the antigen within the demarcation only, giving each pixel the same probability of becoming the location of a receptor while keeping a minimum distance of 6 nm between receptors. In the 2nd step, the simulated particle is placed at a random distance (0–30 nm) from the receptor at a random angle. Simulated gold particles cannot be closer than 10 nm to each other or to real gold particles of the other size. Simulated particles (~10% in outer rim) closely mimic the proportion of real GluN1 particles in the outer rim. Fitting of NNDs to keep the spatial relationship between simulated particles similar to that of real ones was performed as described previously [32], except for using the two-step method when relocating a particle.

Simulating particle distributions outside of exclusion zones was performed similarly as described in [59]: We first randomly distributed 1.4 + 226∗AZ area (rounded to the nearest integer) exclusion zone centers with a minimum distance of 40 nm from each other within the AZ. Exclusion zones had a diameter of 50 nm. We then performed two-step simulations as described above with the modification that simulated antigens were not allowed within exclusion zones. Gold particles simulated in the 2nd step were allowed within exclusion zones. Each simulation was performed 500 times and average values for each synapse were computed. 

#### 4.5.4. Center Periphery Index (CPI)

To compute the CPI, we first computed the normalized distance from center as distance from the center of gravity divided by sum of distance from center of gravity and distance from nearest demarcation border. Animal means were calculated on this normalized distance from center. The mean normalized distance from center of particles randomly distributed in a circle corresponds to the radius of the inner circle (r_ic_) which has half of the area of the full circle. In the unit circle (r_uc_ = 1) r_ic_ can be calculated as follows:$$r_{ic}^{2}\pi \;=\;\frac{r_{uc}^{2}\pi }{2}$$
$$r_{ic}^{2}\pi \;=\;\frac{\pi }{2}$$
$$r_{ic}\;=\;\frac{1}{\sqrt{2}}\approx 0.707$$

The CPI is calculated as the square of the mean distance from center. For particles randomly distributed in a circle, the CPI is ≈ 0.5, aiding the intuitive interpretability of the metric. For synapses, which are usually not completely circular, we typically observe CPIs between 0.5 and 0.6 for randomly distributed particles.

#### 4.5.5. Particle Cluster Analysis

Particle cluster analysis was performed as described previously [32]. Briefly, a particle cluster was defined by two properties: The minimum number of particles necessary to form a cluster and a maximum distance. Particles closer than this distance will belong to the same cluster. We set the minimum number of particles to 3 and the maximum distance between particles to the mean NND plus two times its standard deviation. For Cav2.1 particle clusters shown in Figure 4, the maximum distance was 39.5 nm (21.8 + 2 × 8.85 nm).

### 4.6. Serial Section EM Analysis of PSD Area

Mice were deeply anaesthetized with Ketamine/Xylazine (100/10 mg/kg BM, i.p.) and perfused transcardially with 4% PFA, 0.05% glutaraldehyde, 15% saturated picric acid in 0.1M PB, pH 7.3–7.4. The brain was extracted and 50 µm coronal slices were cut with a vibratome (VT1000S, Leica, Wetzlar, Germany). Slices containing the dorsal hippocampus were stained with 1% OsO_4_ and contrasted with 1% uranyl acetate, followed by gradual dehydration with EtOH, and the flat-embedded in Durcupan resin. Serial ultrathin sections sized 70 nm were then cut from the middle one-third of SR with an ultramicrotome (UC7, Leica, Wetzlar, Germany). Serial sections were imaged with a Tecnai 10 or 12 electron microscope. PSD length was measured on each serial image and multiplied with the section thickness to obtain the PSD area.

### 4.7. Statistics and Visualization

Analytic statistics were performed in Prism 8 (GraphPad, San Diego, CA, USA) except for paired *t*-tests comparing NNDs between real and simulated particles which were performed by Darea and followed by Holm–Sidak multiple comparison correction performed in Prism 8. Example EM images with annotated demarcation and/or gold particles were prepared with Darea software. Graphs were prepared in Prism 8. Error bars indicate the standard error unless described otherwise. To indicate *p*-value in graphs, we use # *p* < 0.1, * *p* < 0.05, ** *p* < 0.01, *** *p* < 0.001. For the sake of readability, we report only *p*-values rounded to two significant digits in the main text and figure captions. Details on the statistics including F and *t* statistics can be found in Appendix A.

## Figures and Tables

**Figure 1 ijms-21-06737-f001:**
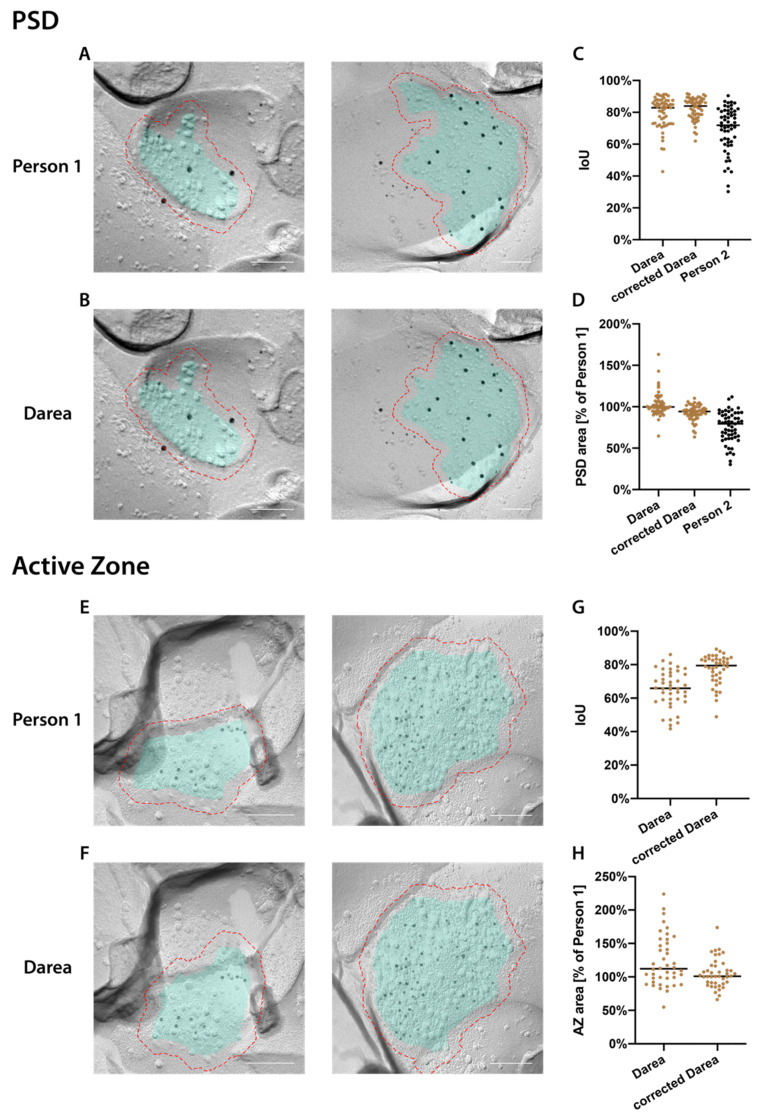
Automated demarcation by Darea. (**A,B**): Representative images of manual (**A**) and machine-demarcated (**B**) postsynaptic density (PSD) labeled for α-amino-3-hydroxy-5-methyl-4-isoxazolepropionic acid (AMPA) receptors (AMPAR) (6 nm gold particles) and GluN1 (12 nm gold particles) in the CA1 area. Cyan overlay indicates demarcated areas. Red dashed line indicates 30 nm rim around the demarcation. (**C,D**): Comparison of the accuracy of automated and human demarcation, calculated as the intersect over union (IoU) between the two demarcations (**C**) or percentage of demarcated PSD area (**D**) by Darea, Darea followed by human correction (corrected Darea) or manual demarcation by Person 2 with respect to manual demarcation by Person 1. Horizontal bars indicate the median. (**E,F**): Representative images of manual (**E**) and machine-demarcated (**F**) AZ labeled for Cav2.1 (5 nm gold particles) in the CA1 area. (**G,H**): Accuracy of automated demarcation shown as IoU (**G**) or percentage of demarcated AZ area (**H**) by Darea or by Darea followed by human correction (corrected Darea) with respect to manual demarcation. Horizontal bars indicate the median. Scale bars: 100 nm.

**Figure 2 ijms-21-06737-f002:**
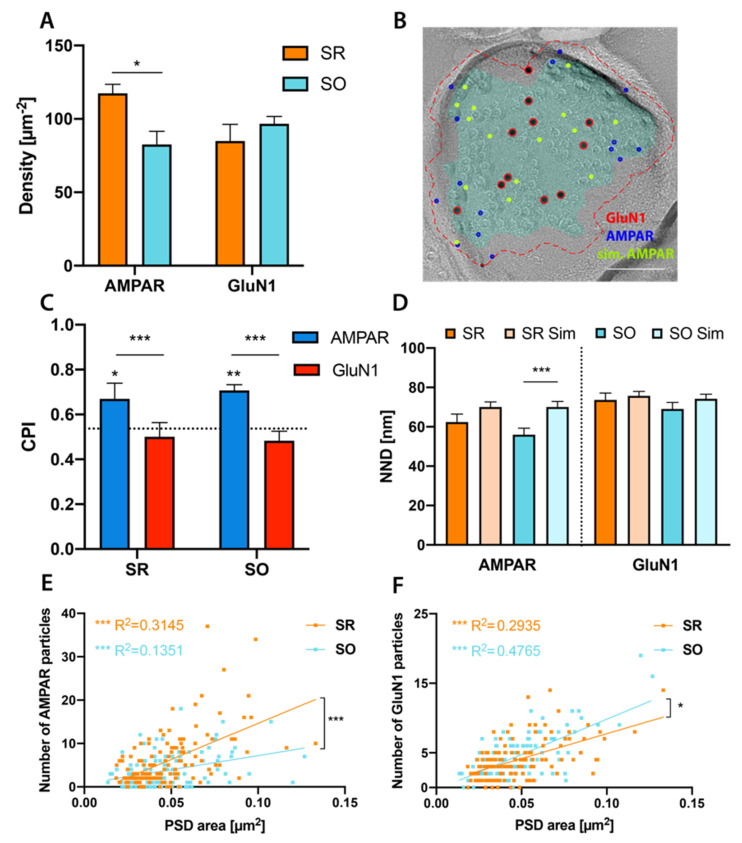
Glutamate receptor distribution in stratum radiatum (SR) and stratum oriens (SO) PSD. (**A**): Labeling density of GluA1-3 AMPAR (6 nm) and GluN1 (12 nm). AMPAR labeling is significantly denser in SR than SO (SR: 118 µm^−2^, SO: 83 µm^−2^, *p* = 0.018) while GluN1 labeling density is similar in both strata (SR: 85 µm^−2^, SO: 97 µm^−2^, *p* = 0.49) (*n* = 4 mice, paired *t*-test with Holm–Sidak multiple comparison correction). (**B**): Example image showing Monte-Carlo simulation of randomly distributed AMPAR particles. Red, blue and green circles indicate real GluN1, real AMPAR and simulated AMPAR particles, respectively. Cyan overlay indicates the demarcation of PSD, red dashed line indicates outer rim 30 nm from the demarcation. Scale bar: 100 nm. (**C**): Center-Periphery index (CPI) of AMPAR and GluN1 particles in SR and SO. Dotted line at 0.536 indicates CPI of randomly distributed particles. In the example image shown in (**B**), the CPI for AMPAR and GluN1 particles is 0.95 and 0.42, respectively. GluN1 particles are distributed similar to random distribution in both strata (SR: *p* = 0.97, SO: *p* = 0.75). AMPAR particles in SR and SO are distributed significantly more peripheral than expected by chance (SR: *p* = 0.029, SO: *p* = 0.0059), as well as more peripheral than GluN1 particles (SR: *p* = 0.0007, SO: *p* = 0.0001) (*n* = 4 mice, 3-way analysis of variance (ANOVA) with Sidak’s multiple comparison test; 225 partial and 30 complete PSDs were included in this analysis). (**D**): Comparison of real and simulated nearest-neighbour differences (NNDs) between AMPAR or GluN1 particles. Real NNDs are significantly smaller than simulations for AMPAR in SO (*p* = 0.0004, *n* = 76 synapses) but not in SR (*p* = 0.13, *n* = 113 synapses). In contrast, real and simulated NNDs between GluN1 are similar in both strata (SR: *p* = 0.4, *n* = 110 synapses; SO: *p* = 0.13, *n* = 107 synapses) (paired *t*-test followed by Holm–Sidak multiple comparison correction). (**E,F**): Significant positive correlation of number of AMPAR (**E**) or GluN1 (**F**) particles with synapse size (AMPAR: SR: R^2^ = 0.3145, *p* < 0.0001, SO: R^2^ = 0.1351, *p* < 0.0001; GluN1: SR: R^2^ = 0.2935, *p* < 0.0001, SO: R^2^ = 0.4765, *p* < 0.0001, linear regression followed by Holm–Sidak multiple comparison correction). The regression line was significantly steeper for AMPAR in SR than in SO (F = 14.51, *p* = 0.0002) and for GluN1 in SO than in SR (F = 4.172, *p* = 0.031) (SR: *n* = 135 synapses, SO: *n* = 120 synapses from 4 mice). * *p* < 0.05, ** *p* < 0.01, *** *p* < 0.001.

**Figure 3 ijms-21-06737-f003:**
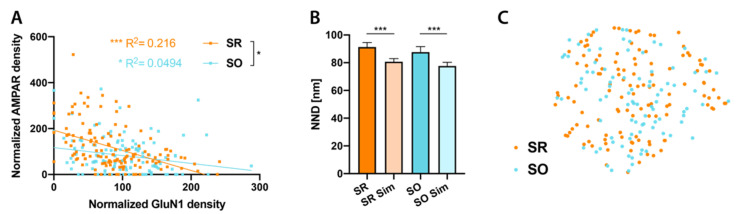
Relationship between AMPAR and N-methyl-D-aspartate (NMDA) receptor (NMDAR) distributions. (**A**): Significant negative correlation between normalized GluN1 and AMPAR particle densities in both SR and SO (SR: R^2^ = 0.216, *p* < 0.0001; SO: R^2^ = 0.0494, *p* = 0.019, linear regression followed by Holm–Sidak multiple comparison correction). Regression line is significantly steeper in SR than SO (F = 5.12, *p* = 0.021). (**B**): In both SR and SO, NNDs between real AMPAR particles and real GluN1 particles are significantly larger than those between real AMPAR particles and simulated GluN1 particles (SR: *p* < 0.0001, *n* = 119 synapses; SO: *p* = 0.0002, *n* = 92 synapses), indicating subsynaptic segregation of AMPA and NMDA receptors (paired *t*-test followed by Holm–Sidak multiple comparison correction). **C**: *t*-distributed stochastic neighbor embedding (*t*SNE) plot of PSDs with labeling for both AMPAR and GluN1 showing SO and SR synapses. * *p* < 0.05, *** *p* < 0.001.

**Figure 4 ijms-21-06737-f004:**
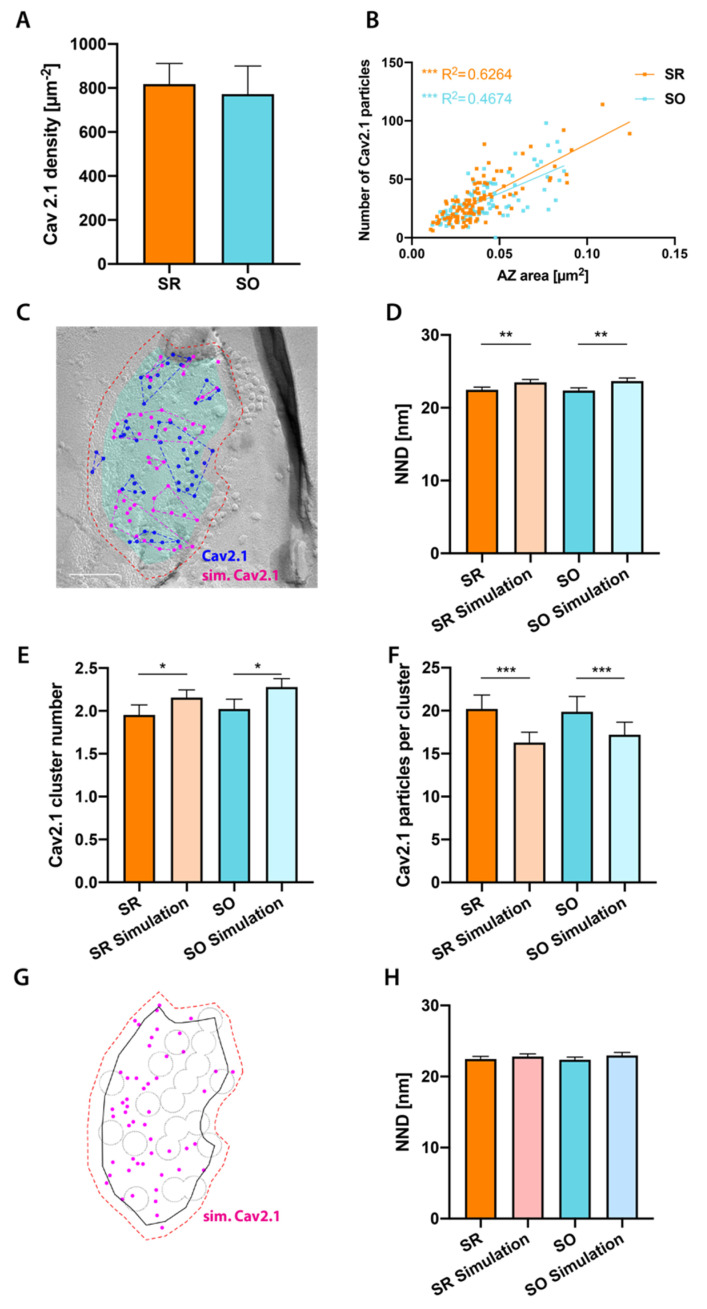
Cav2.1 channel distribution is similar in SR and SO active zones. (**A**): Density of Cav2.1 labeling with 5 nm gold particles is similar in SR and SO (SR: 818 µm^−2^, SO: 772 µm^−2^, *p* = 0.54, *n* = 4 mice, paired *t*-test, *t*_(3)_ = 0.679). (**B**): Significant positive correlation of Cav2.1 particle number with active zone (AZ) size in both layers (SR: R^2^ = 0.6264, *p* < 0.0001, *n* = 109 synapses; SO: R^2^ = 0.4674, *p* < 0.0001, *n* = 95 synapses, linear regression followed by Holm–Sidak multiple comparison correction). Regression line is similarly steep in SR than SO (F = 2.223, *p* = 0.1375). (**C**): Example image showing Monte-Carlo simulation of randomly distributed Cav2.1 particles. The CPI of Cav2.1 in this image is 0.57. Cyan overlay indicates the demarcation of AZ, red dashed line indicates outer rim 30 nm from the demarcation. Blue and magenta circles indicate real and simulated Cav2.1 particles, respectively. Blue and magenta dashed lines indicate real and simulated Cav2.1 clusters, respectively. Scale bar: 100 nm. (**D**): NNDs between real Cav2.1 particles are significantly smaller than between Monte-Carlo simulated Cav2.1 particles in both strata (SR: *p* = 0.0046, *n* = 109 synapses; SO: *p* = 0.0059, *n* = 94 synapses, paired *t*-test followed by Holm–Sidak multiple comparison correction). (**E**): The number of Cav2.1 clusters per AZ is significantly smaller in real than simulated Cav2.1 particles in both strata (SR: *p* = 0.012, *n* = 109 synapses; SO: *p* = 0.012, *n* = 95 synapses, paired *t*-test followed by Holm–Sidak multiple comparison correction). (**F**): Number of Cav2.1 particles per cluster is significantly higher than in Monte-Carlo simulations in both strata (SR: *p* = 0.0004, *n* = 109 synapses; SO: *p* = 0.0003, *n* = 95 synapses, paired *t*-test followed by Holm–Sidak multiple comparison correction). (**G**): Example of exclusion zone simulation in the same synapse as shown in panel (**C**). Solid black line indicates demarcation, red dashed line indicates outer rim, black dotted lines indicate exclusion zones and magenta dots indicate simulated Cav2.1 particles. (**H**): NNDs between real Cav2.1 particles are similar to those between simulated Cav2.1 particles based on the exclusion zone model in both strata (SR: *p* = 0.29, *n* = 109 synapses; SO: *p* = 0.19, *n* = 94 synapses, paired *t*-test). * *p* < 0.05, ** *p* < 0.01, *** *p* < 0.001.

**Table 1 ijms-21-06737-t001:** Primary and secondary antibodies used in this study.

Antibody	Reference	Concentration or Dilution
Rb-anti-GluA1-3	[45]	4.3 µg/mL
Ms-anti-GluN1	Millipore, Burlington, MA, USA, Cat.No. MAB363	5.63 µg/mL
Gp-anti-Cav2.1	Synaptic Systems, Göttingen, Germany, Cat.No. 152205	2.5 µg/mL
Gt-anti-ms-6nm	Jackson ImmunoResearch, West Grove, PA, USA, Cat.No. 115-195-146	1:30
Gt-anti-ms-12nm	Jackson ImmunoResearch, Cat.No. 115-205-146	1:30
Gt-anti-rb-6nm	Jackson ImmunoResearch, Cat.No. 111-195-144	1:30
Gt-anti-rb-12nm	Jackson ImmunoResearch, Cat.No. 111-205-144	1:30
Gt-anti-rb-5nm	BBI Solutions, Crumlin, UK, Cat.No. EM.GAR5	1:30
Gt-anti-gp-5nm	BBI Solutions, Cat.No. EM.GAG5	1:50

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
