# Peer review of "Deep Learning-Assisted High-Throughput Analysis of Freeze-Fracture Replica Images Applied to Glutamate Receptors and Calcium Channels at Hippocampal Synapses"

_ijms, 2020, doi:10.3390/ijms21186737_

Round 1
Reviewer 1 Report
Authors newly developed Darea software for deep learning of replica images, which enables demonstrate gold particles of pre- and postsynapses in a reproducible manner.
The authors investigated glutamate receptors (AMPA and NMDA receptors) and calcium channel(Cav2.1) distributions between hippocampal CA3-CA1 spine synapses and found the difference in the density of AMPA and NMDA receptors at apical (in SR) and basal (in SO) synapses, with their fine freeze-fracture replica labeling. At the presynaptic sites, they found that Cav2.1 channels show similar densities and cluster sizes in apical and basal synapses with distributions. And they concluded that their analysis supports an exclusion zone model, with Cav2.1 being excluded from vesicle release sites.
This manuscript provides us with the basic information of synapses in the hippocampal CA1 pyramidal using their newly-developed Darea for automatically demarcating the PSD and detecting gold particles, and the data quality is good. I believe the Darea software will prompt big data science in this field. Although there are a few concerns about this manuscript (e.g., data display and discussion), I believe it must be good enough to be accepted to the journal after revision.
For accepting the manuscript, I am sorry that their discussion is not persuasive. And, the explanation of how they develop Darea is less. “4.5. Darea software” with four-line is not understandable and suitable, considering the title.
Please give adequate responses to my comments below.
Comments:
- Page 2, Line 9,
… and is affected by disease is a central challenge in molecular …
-> and is affected by diseases are central challenges in molecular …
- Page 3, Line 19,
Please specify “(DL)” in the text and Abbreviations in page16.
- Page 4, Figure 1,
/ In A-B top & E-F top, please correct position of “PSD” and “Active Zone”, respectively.
/ Specify horizontal bars in C, D, G, and H.
- Page 6, Figure 2,
/ Please provide bar graphs pf the “particle numbers per synapse”, associated with the current Figure2A.
/ Figure 2D and all the following bar graphs in this manuscript: Please don’t use the texture appearance of the bars (i.e., SR Sim & SO Sim of Fig 2D). I recommend filled and blank, or other styles.
/ In E & F bottom, please correct position of “PSD area [um2]”.
/ “PSD Area [um2]” -> “PSD area [um2]” (Need the capital correction)
- Page 7, Figure 3,
/ On the left side and bottom of Fig 3A, please change the label “normalized” to that with a capital N.
/ Specify horizontal bars in C, D, G, and H.
- Page 8, Line 7,
/ with few notable exceptions -> with a few notable exceptions
- Page 8, Line 23,
/ [41] I cannot understand this citation. This paper seems not to be a paper about the silent synapses. It is Drosophila EM and zebrafish EM study. Don’t you intend to mention about silent papers of mature mice?
- Page 8, Line 24,
“This result is in line with our previous study showing that all cerebellar parallel fiber to Purkinje cell synapses have AMPAR labeling [31], which is apparently inconsistent with electrophysiological results [51].”
/ [51] I cannot interpret this citation.
Isope & Barbour (2002)[51] suggested a very small connection probability by paired-recording of unitary Granule Cell→Purkinje Cell synapses. Comparing to your previous result in which all Purkinje cell synapses have AMPARs [31], the results of Isope & Barbour (2002) has been utterly controversial. This point is understandable. However, a recent somato-dendritic recording study by Ohtsuki (2020, J Neurosci; doi.org/10.1523/JNEUROSCI.3211-18.2019) showed that the EPSC generated in the distal dendrites do not transmit to the soma efficiently because of the low electroconductivity, implying that the results of Isope & Barbour (2002) have underestimation due to their recording solely from soma. Therefore, your concern about the divergence between the EM study and electrophysiology in cerebellar Purkinje cells would be essentially solved.
- Page 10, Figure 4,
/ Please provide bar graphs the “particle numbers per AZ”, associated to the current Figure 4A.
/ In B bottom, please correct the position of “AZ Area [um2]”.
/ “AZ Area [um2]” -> “AZ area [um2]” (Need the capital correction)
/ Other labels in A and E need capital correction, too.
/ Associated with the current Figure 4D-F, please provide bar graphs # of docked vesicles per AZ.
- Legends of Figure 4 C (Page 11, Lines 6-10): This is not enough for the explanation of your clustering-method. The authors must describe the exact method for the clustering somewhere.
- Page 11, Line 37,
/ 100 images), but
: Notice the cyan color of comma, please.
- Page 12, Line 1-4: “In acute slice experiments the acute phase of LTP (up to a few hours) is observed. In vivo, however, different long-term regulatory mechanisms (days or weeks), such as homeostatic synaptic scaling [59] and structural synaptic plasticity [60] may lead to the state we observed.
/ Please rewrite the sentences. What is the “acute” phase?
- Page 12, Line 4,
/ “… explained by GluA1 incorporation” -> “… explained by activity-dependent GluA1 incorporation”
- Page 12, Line 8-9: “where large synapses have been shown unable to undergo LTP in organotypic slice culture [62].”
/ I don’t think this organotypic culture study is directly related to your finding, while not disagreeing the structural plasticity of spines.
- Pages 11-12, Line 48-19 (next page),
/ These sections are not organized. Please rewrite it.
/ Regarding your finding of Figure 3A, are AMPA/NMDA ratios different from by the SR- and SO-stimulations?
/ Location-dependent difference in AMPAR expression of CA1 pyramidal cell dendrites is studied previously (Nicholson DA, et al., 2006, Neuron, DOI 10.1016/j.neuron.2006.03.022). Please give a mention of that. And, if possible, please compare it to your results.
/ Personally, I would like to know such differences in the expression of glutamate-receptor subtypes along the dendrites and branches in biopsy or samples of other species, like human beings or primates. The audience may also want to know it. Can you give a mention on that?
- Pages 14, Line 28,
/ [32] Please describe how you generate the random number (i.e., algorithm).
- Pages 14, Lines 45-46: “Each simulation was performed 50 times, and average values for each synapse were computed.”
/ In [32], you did sampling 500 times. 50 times sounds too small or even be unacceptable.
- Pages 17, “References” (, but not Reference, probably) seem not completed (e.g., [32]). Please double-check it.
- Supplemental Figures: In Figure S1B, what is the “Human”, but not “Person X”?
- Supplemental Figure Legends: In Figure S4, the title “Figure S4. AMPAR single labeling reproduces results from Fig. 2” is not adequate. Please reconsider it.
/ I could not find obvious other errors and typos this time.
Author Response
Response to Reviewer 1 Comments
Point 0: “4.5. Darea software” with four-line is not understandable and suitable, considering the title.
Response 0: We have added more details to this section, and would like to point out that the further details of the different parts of Darea software can be found in the following subsections 4.5.1 – 4.5.5.
Point 1: 1. Page 2, Line 9,
… and is affected by disease is a central challenge in molecular …
-> and is affected by diseases are central challenges in molecular …
Response 1: We have corrected accordingly.
Point 2: 2. Page 3, Line 19,
Please specify “(DL)” in the text and Abbreviations in page16.
Response 2: We have changed DL to Darea to improve the consistency and clarity.
Point 3: 3. Page 4, Figure 1,
/ In A-B top & E-F top, please correct position of “PSD” and “Active Zone”, respectively.
/ Specify horizontal bars in C, D, G, and H.
Response 3: Thank you for pointing out these issues. We modified the position of the axis titles and specified that the horizontal bars indicate the median.
Point 4: 4. Page 6, Figure 2,
/ Please provide bar graphs pf the “particle numbers per synapse”, associated with the current Figure2A.
/ Figure 2D and all the following bar graphs in this manuscript: Please don’t use the texture appearance of the bars (i.e., SR Sim & SO Sim of Fig 2D). I recommend filled and blank, or other styles.
/ In E & F bottom, please correct position of “PSD area [um2].
/ “PSD Area [um2]” -> “PSD area [um2]” (Need the capital correction)
Response 4: Unfortunately, most of the PSDs we analysed are only partially fractured, with only 17 (13% of total PSDs) and 13 (11%) complete ones for SR and SO, respectively. Including partially fractured synapses in the analysis will not (or minimally) affect the particle density values but would lead to a strong underestimation of particle number per synapse. Giving a reliable estimate for particle numbers per PSD based on only complete PSDs is also not possible due to the small number of complete PSDs we observed and a potential bias toward smaller synapses that can be fully fractured with greater chance. We therefore refrained from including such a graph in the manuscript.
We modified the bar graphs and now use darker and lighter colors for real and simulated data points.
We corrected the position and capital.
Point 5: 5. Page 7, Figure 3,
/ On the left side and bottom of Fig 3A, please change the label “normalized” to that with a capital N.
Response 5: Corrected accordingly.
Point 6: 6. Page 8, Line 7,
/ with few notable exceptions -> with a few notable exceptions
Response 6: Corrected accordingly.
Point 7: “7. Page 8, Line 23,
/ [41] I cannot understand this citation. This paper seems not to be a paper about the silent synapses. It is Drosophila EM and zebrafish EM study. Don’t you intend to mention about silent papers of mature mice?
Response 7: We added the wrong citation here, thank you very much for pointing this out. We changed it to the correct citation.
Point 8: 8. Page 8, Line 24,
“This result is in line with our previous study showing that all cerebellar parallel fiber to Purkinje cell synapses have AMPAR labeling [31], which is apparently inconsistent with electrophysiological results [51].”
/ [51] I cannot interpret this citation.
Isope & Barbour (2002)[51] suggested a very small connection probability by paired-recording of unitary Granule Cell→Purkinje Cell synapses. Comparing to your previous result in which all Purkinje cell synapses have AMPARs [31], the results of Isope & Barbour (2002) has been utterly controversial. This point is understandable. However, a recent somato-dendritic recording study by Ohtsuki (2020, J Neurosci; doi.org/10.1523/JNEUROSCI.3211-18.2019) showed that the EPSC generated in the distal dendrites do not transmit to the soma efficiently because of the low electroconductivity, implying that the results of Isope & Barbour (2002) have underestimation due to their recording solely from soma. Therefore, your concern about the divergence between the EM study and electrophysiology in cerebellar Purkinje cells would be essentially solved.”
Response 8: Thank you for pointing out this paper, of which we were not aware. We agree that it would explain the apparent inconsistency between EM and the electrophysiological results. We have removed this part about the cerebellum and added some discussion about silent synapses in the hippocampus.
Point 9: 9. Page 10, Figure 4,
/ Please provide bar graphs the “particle numbers per AZ”, associated to the current Figure 4A.
/ In B bottom, please correct the position of “AZ Area [um2]”.
/ “AZ Area [um2]” -> “AZ area [um2]” (Need the capital correction)
/ Other labels in A and E need capital correction, too.”
/ Associated with the current Figure 4D-F, please provide bar graphs # of docked vesicles per AZ.
Response 9: Unfortunately, most of the AZs we analyzed are partially fractured ones, similar to the PSDs. While this does not affect the values for Cav2.1 density, it would greatly affect the values for Cav2.1 particle numbers per AZ. If we only use completely fractured AZs, for two animals, we do not have enough number of complete AZs to give a reliable estimate or conduct statistics. For this reason, we did not include such a graph.
We corrected the position and capital.
The number of docked vesicles we used for comparison are from Schikorski and Stevens, 1997. We did not conduct measurements of docked vesicle numbers ourselves and can therefore not provide bar graphs. We now refer to the exact figure in Schikorski and Stevens, 1997, from which we got the docked vesicle data.
Point 10: 10. Legends of Figure 4 C (Page 11, Lines 6-10): This is not enough for the explanation of your clustering-method. The authors must describe the exact method for the clustering somewhere.
Response 10: We have added the explanation of our clustering method in a new subsection 4.5.5 in Materials and Methods.
Point 11: 11. Page 11, Line 37,
/ 100 images), but
: Notice the cyan color of comma, please.”
Response 11: Corrected.
Point 12: 12. Page 12, Line 1-4: “In acute slice experiments the acute phase of LTP (up to a few hours) is observed. In vivo, however, different long-term regulatory mechanisms (days or weeks), such as homeostatic synaptic scaling [59] and structural synaptic plasticity [60] may lead to the state we observed.
/ Please rewrite the sentences. What is the “acute” phase?
Point 13: 13. Page 12, Line 4,
/ “… explained by GluA1 incorporation” -> “… explained by activity-dependent GluA1 incorporation”
Page 12, Line 8-9: “where large synapses have been shown unable to undergo LTP in organotypic slice culture [62].”
/ I don’t think this organotypic culture study is directly related to your finding, while not disagreeing the structural plasticity of spines.
Point 14-1: Pages 11-12, Line 48-19 (next page),
/ These sections are not organized. Please rewrite it.
Response 12-14: We have rewritten this section taking into account these comments.
Point 14-2: / Regarding your finding of Figure 3A, are AMPA/NMDA ratios different from by the SR- and SO-stimulations?
Response 14-2: Unfortunately, we could not find any literature reporting the difference in AMPA/NMDA ratios between SR- and SO-stimulations.
Point 14-3: / Location-dependent difference in AMPAR expression of CA1 pyramidal cell dendrites is studied previously (Nicholson DA, et al., 2006, Neuron, DOI 10.1016/j.neuron.2006.03.022). Please give a mention of that. And, if possible, please compare it to your results.
Response 14-3: Due to this known gradient of AMPAR expression, we limited our analysis to the middle one-third of the stratum radiatum in order to avoid biases caused by different distances from the pyramidal cell bodies. We now make a mention of this publication and explicitly explain this point in the result. As we always sample from the same approximate distance, we cannot directly compare our results with the data shown in Nicholson et al.
Point 14-4: / Personally, I would like to know such differences in the expression of glutamate-receptor subtypes along the dendrites and branches in biopsy or samples of other species, like human beings or primates. The audience may also want to know it. Can you give a mention on that?
Response 14-4: In a different project in our group, we have examined the density gradient of synaptic GluA1, GluA2 and GluA3 subunit expression along the pyramidal cell dendrites in the CA1 area of the mouse, showing higher density for GluA2, but not GluA1 and GluA3, toward distal dendrites in the stratum radiatum (Jevtic et al, under preparation). However, similar analysis in primates would be technically difficult because pyramidal cells in the primate CA1 area are dispersed across the layer, making the distance measurement not feasible in replica samples.
Point 15: 15. Pages 14, Line 28,
/ [32] Please describe how you generate the random number (i.e., algorithm).”
Response 15: We have provided details on the simulation, and mentioned the algorithm we used.
Point 16: / In [32], you did sampling 500 times. 50 times sounds too small or even be unacceptable.
Response 16: We have redone our analysis with 500 simulations as requested. We found subtle changes in the simulated values and p-values, but it did not change any of our previous conclusions. We have updated figures, figure captions, appendix, and p-values in the text accordingly.
Point 17: 17. Pages 17, “References” (, but not Reference, probably) seem not completed (e.g., [32]). Please double-check it.
Response 17: Thank you for pointing this out. We have checked the references and corrected where appropriate.
Point 18: 18. Supplemental Figures: In Figure S1B, what is the “Human”, but not “Person X”?
Response 18: We have changed the label to “Person 1”.
Point 19: 19. Supplemental Figure Legends: In Figure S4, the title “Figure S4. AMPAR single labeling reproduces results from Fig. 2” is not adequate. Please reconsider it.
Response 19: We have changed the figure title to “Figure S4. AMPAR distribution examined by single labeling with 5 nm gold particles.”
Reviewer 2 Report
This lab has an excellent track record in producing gorgeous immunogold labeled F-F replicas of synapses. It has always been a joy to see these beautiful images of en face view of the synaptic membranes with specific labeling of different proteins. The authors present here a new method of “deep learning-assisted high-throughput analysis” to save substantial amount of time in demarcating area of interest on replicas by semi-automation, and presents data on select glutamate receptors on the postsynaptic membrane, and calcium channels on presynaptic membranes. This study focuses on the well-studied CA3 to CA1 apical synapses in the middle segment of the stratum radiatum (SR), and uses the synapses in the stratum oriens (SO) for comparison.
The introduction is well laid out and referenced, and the results and analysis are thorough as always. There are only minor questions and concerns:
- P3, line 19, a few words explaining why DL is standing for “machine generated” would be nice.
- This reviewer really appreciates the additional images included in Fig. S1, and noticed that the errors by DL are mostly through jumping fracture planes. Although this is easily corrected manually, is there any way to program for avoiding this error?
- P5 line 12-14, “finding more AMPA R than NR in the 30 nm rim” could be due to the much higher presence of extrasynaptic AMPARs than NRs. Extrasynaptic receptors were discussed in this lab’s previous work, and this point should be discussed here as it could very well affect the interpretation that “AMPAR is significantly more peripherally located while NR is evenly distributed”.
- Although the reason for including this 30 nm rim in the counting of label was explained in the text, this reviewer finds that including 30 nm may be a bit over generous. I am curious as to what would happen if the 30 nm rim is removed from data? Would the CPI values substantially change, especially for the AMPAR? Especially with the data from the 5 nm gold labeling samples.
- While Fig. 2B is a nice illustration, it would help the readers to get a quick idea of what the CPI stand for if the CPI values for the different colored labels in this example are included in the figure legend.
- 8, line 4, The exp design to switch the difference sized gold for the different ab’s is very carefully thought out, but why are these data in supplementary (Fig. S3)? Are they not equally important?
- P8, line 10, if results from this set was presented as “exceptions”, then there should be a “third” point at the beginning of line 10 following the first and second points.
- The figure legend of S4 should spell out at the beginning that these are 5 nm 2nd ab results, and please include a representative EM image if possible.
- P8, line 35-36, comparing area between complete AZ from replica and reconstructed PSD from serial sections (panel B of Fig. S5), how many AZ and PSD were analyzed? The sample size information is missing from the figure legend.
- One side remark for authors’ consideration– based on images of thin sectioned PSD shown in Fig. S5, the PSD is relatively thick. Both the thickness and curvature of this PSD suggest that this synapse is under an excitatory state, perhaps due to perfusion fixation. This might explain the lack of “silent synapse” in these samples. Since the discrepancy on “silent synapses” between electrophysiology recording and F-F immunogold labeling has been discussed in this group’s previous reports, it may not be necessary to mention it here because it was not further discussed in the present manuscript, and it is not a central point of this paper. Especially in view of the statement on P14 line 29, in the methods, “synapses containing at least 2 gold particles were included for analysis”. Does this mean that there are synapses containing less than 2 particles, and could these be “silent synapses”?
- P16, Abbreviations list, DL, IoU, are missing from the list.
Author Response
Response to Reviewer 2 Comments
Point 1: 1. P3, line 19, a few words explaining why DL is standing for “machine generated” would be nice.
Response 1: We have changed “DL” to “Darea”.
Point 2: 2. This reviewer really appreciates the additional images included in Fig. S1, and noticed that the errors by DL are mostly through jumping fracture planes. Although this is easily corrected manually, is there any way to program for avoiding this error?
Response 2: We agree with your observation this jumping through fracture planes is a typical scenario where Darea’s demarcation is quite different from what it should be. However, this occurs only in a small fraction of images, even when only considering those where such a jumping fracture plane exists. An algorithm for automatically correcting this needs to have a very small false positive rate (i.e. the number of images, where it corrects although it should not, needs to be very small), otherwise it could easily do more harm than good. We therefore think writing such an algorithm would be a quite challenging task. However, we think that through further increasing the number of training images, Darea will eventually learn to avoid such mistakes by itself.
Point 3: 3. P5 line 12-14, “finding more AMPA R than NR in the 30 nm rim” could be due to the much higher presence of extrasynaptic AMPARs than NRs. Extrasynaptic receptors were discussed in this lab’s previous work, and this point should be discussed here as it could very well affect the interpretation that “AMPAR is significantly more peripherally located while NR is evenly distributed
Response 3: We agree that the choice of the outer rim size is a relevant parameter, which affects the resulting CPI and can therefore easily affect the conclusions. Too large outer rim would result in overestimation of the CPI, particularly for AMPAR, due to extrasynaptic receptors. Too small outer rim would underestimate the CPI because some particles representing receptors within the PSD but close to the edge of the synapse would be missed in the analysis. To verify our selection of 30 nm width for the outer rim, we quantified the average number of particles in the outer rim for the 5 nm AMPAR labeling with different outer rim sizes (new supplementary figure S3). We find a steep linear increase in the number of particles up to a rim size of 30 nm. With rim sizes larger than that, we see a much less steep increase. We interpret this as the steep increase up to 30 nm stemming from particles belonging to epitopes located within the PSD and the shallower increase stemming from extrasynaptic AMPARs. We therefore think the main contribution to the counted labeling in the 30 nm outer rim should come from receptors within the PSDs, with extrasynaptic AMPARs only accounting for a minor population.
Point 4: 4. Although the reason for including this 30 nm rim in the counting of label was explained in the text, this reviewer finds that including 30 nm may be a bit over generous. I am curious as to what would happen if the 30 nm rim is removed from data? Would the CPI values substantially change, especially for the AMPAR? Especially with the data from the 5 nm gold labeling samples.
Response 4: We quantified the CPI without the outer rim and also conducted simulations relocalizing the gold particles only within the PSD for GluN1/AMPAR double labeling and 5 nm AMPAR single labeling (see graphs below). GluN1 particles were still significantly more central than AMPAR particles, however, now GluN1 showed a significant preference for the center of the PSD while AMPAR showed a similar distribution as the simulation in SR and preference for periphery in SO. In case of 5 nm labeling for AMPAR, there is no preference for periphery or center in both strata.
Figure 1. CPI of 6 nm AMPAR 12 nm GluN1 labeling (A) or 5 nm AMPAR labeling (B) when excluding the outer rim. (Please see the attachment)
In case of NMDARs, which have a much lower density of extrasynaptic than synaptic receptors, this difference (random distribution with outer rim, central preference without outer rim) should be caused by excluding particles representing receptors close to the edge, where the particles just happened to be outside of the PSD. Since the same kind of preference switch happened for both NMDARs and AMPARs, we think the same reason should be responsible for AMPARs.
Taken together, we conclude that the error introduced into our analysis by including some particles representing extrasynaptic receptors (using a 30 nm outer rim) is likely much smaller than the error by excluding some particles representing synaptic receptors (using no outer rim).
Point 5: 5. While Fig. 2B is a nice illustration, it would help the readers to get a quick idea of what the CPI stand for if the CPI values for the different colored labels in this example are included in the figure legend.
Response 5: Thank you for this nice suggestion. We have added the CPI values for all example images to their respective figure captions.
Point 6: 6. 8, line 4, The exp design to switch the difference sized gold for the different ab’s is very carefully thought out, but why are these data in supplementary (Fig. S3)? Are they not equally important?
Response 6: We found much lower labeling efficiency for AMPAR with 12 nm gold antibody, which makes some of the results less reliable than the ones shown in Fig.2, and causes some discrepancy between Fig.2 and Fig.S3. We have confirmed that Fig.2 is the more reliable one by single labeling for AMPAR with 5 nm (Fig.S4), reproducing the findings shown in Fig.2. Thus, we think it appropriate to keep those data in supplementary.
Point 7: 7. P8, line 10, if results from this set was presented as “exceptions”, then there should be a “third” point at the beginning of line 10 following the first and second points.
Response 7: We have added “Third” at the appropriate position.
Point 8: 8. The figure legend of S4 should spell out at the beginning that these are 5 nm 2nd ab results, and please include a representative EM image if possible.
Response 8: Thank you for the useful comment. We have added the information about the 5nm gold particles in the figure title (new S5) and added two example EM images to the Figure S5.
Point 9: 9. P8, line 35-36, comparing area between complete AZ from replica and reconstructed PSD from serial sections (panel B of Fig. S5), how many AZ and PSD were analyzed? The sample size information is missing from the figure legend.
Response 9: Thank you for pointing out this missing information. We have added it to the figure caption.
Point 10: 10. One side remark for authors’ consideration – based on images of thin sectioned PSD shown in Fig. S5, the PSD is relatively thick. Both the thickness and curvature of this PSD suggest that this synapse is under an excitatory state, perhaps due to perfusion fixation. This might explain the lack of “silent synapse” in these samples. Since the discrepancy on “silent synapses” between electrophysiology recording and F-F immunogold labeling has been discussed in this group’s previous reports, it may not be necessary to mention it here because it was not further discussed in the present manuscript, and it is not a central point of this paper. Especially in view of the statement on P14 line 29, in the methods, “synapses containing at least 2 gold particles were included for analysis”. Does this mean that there are synapses containing less than 2 particles, and could these be “silent synapses”?
Response 10: Although we are not sure if the PSD morphology indicates an excitatory state, considering also the other reviewer’s comment on the “silent synapses”, we have removed the discussion about the situation in the cerebellum, and added some discussion about silent synapses in the hippocampus.
For our quantification of synapses expressing AMPARs we have included synapses with less than 2 gold particles as well. We have clarified this point in the methods.
Point 11: 11. P16, Abbreviations list, DL, IoU, are missing from the list.
Response 11: We have added IoU to the abbreviations list. We no longer use DL.
